



# The influence of burn severity on dissolved organic carbon concentrations across a stream network differs based on seasonal wetness conditions post-fire

Katie A. Wampler[1], Kevin D. Bladon[1], Allison N. Myers-Pigg[2,3]

[1]Department of Forest Engineering, Resources, and Management, Oregon State University, Corvallis, 97331, USA
[2]Marine and Coastal Research Laboratory, Pacific Northwest National Laboratory, Sequim, 98382, USA
[3]Department of Environmental Sciences, University of Toledo, Toledo, 43606, USA

*Correspondence to*: Katie A. Wampler (katie.wampler@oregonstate.edu)

**Abstract.** Large, high severity wildfires in many regions across the globe have increased concerns about their impacts on
carbon cycling in watersheds. Altered sources of carbon and changes in catchment hydrology after wildfire can lead to shifts in dissolved organic carbon concentrations (DOC) in streams, which can have negative impacts on aquatic ecosystem health and downstream drinking water treatment. Despite its importance, post-fire DOC responses remain relatively unconstrained in the literature, and we lack critical knowledge of how burn severity, landscape elements, and climate interact to affect DOC concentrations. To improve our understanding of the impact of burn severity on DOC concentrations, we measured DOC at
~100 sites across a stream network extending upstream, within, and downstream of a large, high severity wildfire in Oregon, USA. We collected samples across the study sub-basin during four distinct seasonal wetness conditions. We used our high spatial resolution data to develop spatial stream network (SSN) models to predict DOC across the stream network and to improve our understanding of the controls on DOC concentrations. Spatially, we found no obvious wildfire signal—instead we observed a pattern of increasing DOC concentrations from the high elevation headwaters to the sub-basin outlet, while
the mainstem maintained consistently low DOC concentrations. This suggests that effects from large wildfires may be "averaged" out at higher stream orders and larger spatial scales. With our DOC measurements grouped by burn severity group, we observed a significant decrease in the variability of DOC concentrations in the moderate and high burn severity sub-catchments. However, our SSN models were able to predict decreases in DOC concentrations with increases in burn severity across the stream network. Decreases in DOC concentrations were also highly variable across seasonal wetness
conditions, with the greatest (-1.40 to -1.64 mg L$^{-1}$) decrease in the high severity group during the wetting season. Additionally, our models indicated that in all seasons, baseflow index was more influential in predicting DOC concentrations than burn severity, indicating that groundwater discharge can obscure the impacts of wildfire in a stream network. Overall, our results suggest that landscape characteristics can regulate the DOC response to wildfire. Moreover, our results also indicate that the seasonal timing of sampling can influence the observed response of DOC concentrations to wildfire.



## 1 Introduction

Streams play an active role in transporting and processing terrestrial inputs of carbon from the landscape to oceans (Cole et al., 2007). It has been estimated that terrestrial ecosystems deliver between 1.1–5.1 Pg of carbon annually to inland waters (Drake et al., 2018). Thus, landscape-scale disturbances—such as extreme weather events, forest harvesting, insect and pathogen outbreaks, and wildfire—have the potential to substantially impact regional carbon cycling by altering the sources, transport, and processing of terrestrial organic matter (Achat et al., 2015; Amiro et al., 2009; Brando et al., 2019; Chambers et al., 2007; Kurz et al., 2008). In particular, impacts from large, high severity wildfires are of increasing concern due to shifting wildfire regimes, which have led to recent and particularly notable wildfire seasons across many regions of the globe, including the Pacific Northwest (Abatzoglou et al., 2021; de la Barrera et al., 2018; Boer et al., 2020; Dodd et al., 2018; Lagouvardos et al., 2019; Turco et al., 2019). These extreme fire seasons have been primarily attributed to increasing temperatures and longer periods of fire weather due to climate change (Abatzoglou et al., 2018; Duane et al., 2021; Pausas and Keeley, 2021).

Wildfires can alter terrestrial sources of carbon through significant loss and modification of vegetation and organic and shallow mineral soil layers (Johnson et al., 2007; Miesel et al., 2018). While a large portion of this burned material is volatized to the atmosphere as $CO_2$ and other gases, an estimated 1–28 % of the carbon is transformed to pyrogenic organic matter (PyOM) (Forbes et al., 2006; Preston and Schmidt, 2006; Santín et al., 2015). PyOM contains a spectrum of molecules with varying lability based on combustion temperature, characteristics of the burned material, and the formation mechanisms (Masiello, 2004; Wagner et al., 2018). However, there is evidence that this PyOM can impact ecosystem functioning and drinking water treatment (Hohner et al., 2017; Emelko et al., 2011). Since terrestrial ecosystems are a primary source of carbon for inland waters like streams, the wildfire-altered terrestrial carbon stocks can alter the amount and characteristics of dissolved and particulate organic carbon delivered to streams.

Wildfire can also substantially alter catchment hydrology and flowpaths of water through the landscape, further impacting dissolved organic carbon (DOC) transport from the terrestrial landscapes to streams. Specifically, the loss of vegetation post-fire often leads to decreased evapotranspiration (Ma et al., 2020; Nolan et al., 2014; Poon & Kinoshita, 2018) and increased net precipitation (Kusaka et al., 1983; Stoof et al., 2012; Williams et al., 2019). Moreover, wildfires can affect soil physicochemical properties, resulting in increased soil-water repellency, soil sealing, surface crust formation, soil pore clogging, and changes in bulk density due to collapse of soil aggregates, further leading to shifts in hydrologic flowpaths from hillslopes to streams (Balfour et al., 2014; Doerr et al., 2009; Ebel and Moody, 2017; Larson-Nash et al., 2018). These effects on soil hydraulic properties can produce increased surface runoff, lateral flow, or groundwater resulting in increased potential for peak flows and annual water yields (Atwood et al., 2023; Jung et al., 2009; Onda et al., 2008; Rey et al., 2023; Stoof et al., 2014). The post-fire shifts in carbon stocks, hydrologic flow paths, and contact times of carbon with soil, water, and microbes thus have the potential to further influence post-fire DOC concentrations in streams (Olefeldt et al., 2013).



With both direct controls on carbon stocks and indirect controls on DOC movement, the net impact of wildfire on DOC in streams remains poorly constrained. With a loss of vegetation and soil organic matter, one might expect to observe decreased DOC concentrations post-fire. Indeed, there have been observations of decreases in DOC concentrations in permafrost and mountainous regions within the US, ranging from ~11–95 % declines (Betts and Jones, 2009; Caldwell et al., 2020; Chow et al., 2019; Rodríguez-Cardona et al., 2020; Santos et al., 2019). However, with PyOM remaining on the landscape and increased movement of water to streams, there is also the potential for increases in DOC concentrations. This too has been observed, with increases of ~3–10,000 % measured across mediterranean, humid, semi-arid, and subarctic climates (Burton et al., 2016; Caldwell et al., 2020; Chow et al., 2019; Emelko et al., 2011; Harris et al., 2015; Hohner et al., 2016; Oliver et al., 2012; Revchuk and Suffet, 2014; Uzun et al., 2020; Vila-Escalé et al., 2007). Finally, wildfires across the Western US have also resulted in minimal to no impacts on in-stream DOC (Mast and Clow, 2008; Uzun et al., 2020; Wagner et al., 2015). Inconsistency in observed responses may also be due to the spatial variability in DOC through a stream network that is due to the fact that DOC is not inertly transported, which can lead to nonlinear trends across space (Casas-Ruiz et al., 2020; Wollheim et al., 2015). However, the impact of wildfire on the spatial variability across stream networks remains largely unexplored. Overall, the inconsistency of post-fire responses highlights the need to better understand the controls of stream DOC concentrations across burned basins.

In unburned landscapes, in-stream DOC concentrations tend to be tightly related to hydrology. Periods of increased streamflow are often associated with increased DOC concentrations (Butturini and Sabater, 2000; Koehler et al., 2009; Raymond and Saiers, 2010). However, this can also depend on seasonal wetness conditions within the basin, or sub-annual wetting and drying periods. For example, snowmelt or rewetting after long dry periods can cause increases in DOC that peak prior to streamflow (Dawson et al., 2008; Hornberger et al., 1994; Humbert et al., 2015; Lambert et al., 2013). This is often referred to as flushing, where finite pools of DOC are reconnected during the rewetting periods, causing temporary increases in DOC until the source is depleted (Hornberger et al., 1994).

Differences in seasonal wetness conditions can also impact the importance of other basin characteristics in controlling DOC concentrations (Ågren et al., 2007). Other basin characteristics that have been related to DOC concentrations include aridity, where more arid areas have been correlated with higher DOC concentrations (Kerins and Li, 2023). Vegetation type can also control the quality and leachability of the carbon (van den Berg et al., 2012), while elevation can control air and soil temperatures, which have been linked to microbial activity and DOC production in soils (Kalbitz et al., 2000). Subsurface soil properties like texture and organic matter content can influence the amount of carbon that is produced, held in soils, and available to leach, affecting stream DOC concentrations (van den Berg et al., 2012; Futter et al., 2007; Nelson et al., 1992; Wilson and Xenopoulos, 2008). Groundwater sourced from mineral soils or bedrock is usually quite low in DOC (Leenheer et al., 1974) and therefore, higher groundwater contributions may result in lower stream DOC concentrations. Lastly, DOC concentrations have been related to basin area, although the direction of the impact is variable (Ågren et al., 2007;



Mulholland, 1997) with some suggesting that increasing basin area only acts to remove "extreme" measurements (Creed et
al., 2015).

In burned landscapes, additional factors can also influence DOC concentrations. Post-fire DOC concentrations can be
dependent on time since fire (Parham et al., 2013; Rodríguez-Cardona et al., 2020; Santos et al., 2019), area burned
(Rhoades et al., 2019; Uzun et al., 2020; Chow et al., 2019), and fire severity (Santos et al., 2019). In particular, increases in
burn severity have previously been related to decreased DOC concentrations during baseflow conditions (Santos et al.,
2019). However, the same study noted that site-level characteristics were also important in controlling solute responses.
While existing work suggests that burn severity is highly influential on post-fire DOC concentrations, there have been few
studies on this topic and are limited to a small number of sites and burn severity groups (Santos et al., 2019). To improve our
understanding of the impact of burn severity on DOC, a greater range of sites, burn severities, and streamflow conditions
would improve our knowledge of how wildfire effects on DOC propagate through a stream network. Additionally, while
previous work has indicated that site-level characteristics can also be important post-fire, the importance of burn severity
relative to landscape characteristics remains unknown.

In our study, we collected stream water samples to quantify DOC across space and time in a burned sub-basin with a range
of landscape characteristics and burn severities in the Pacific Northwest, USA. We collected water samples across four
distinct hydrologic time periods to answer the following questions: (1) How does DOC vary spatially across a stream
network upstream, within, and downstream of a burn? (2) How do stream DOC concentrations vary within the basin based
on burn severity and antecedent seasonal wetness conditions? (3) What is the relative influence of fire, climate, and
landscape characteristics on DOC concentrations and how do they vary by antecedent seasonal wetness conditions?

## 2 Methods

### 2.1 Study Area

The McKenzie River sub-basin (HUC 17090004) is a 3,461 km$^2$ catchment that is nested in the Willamette River Basin,
which is a tributary of the greater Columbia River basin on the west side of the Cascade Range in Oregon, USA (Fig. 1a).
The sub-basin is the primary source of drinking water for ~200,000 residents near Springfield and Eugene, OR. The upper
two-thirds of the sub-basin are federally owned while the lower one-third is a combination of industrial timberland,
agricultural lands, and private ownership. The sub-basin is ~85 % forested with primarily Douglas-fir (*Pseudotsuga
menziesii*), Western hemlock (*Tsuga heterophylla*), and Pacific silver fir (*Abies amabilis*). The elevation in the sub-basin
ranges from 111 to 3,149 m with a median elevation of 954 m. The mean slope in the sub-basin is ~16 ° with maximum
slopes up to ~50 °. Across the sub-basin, the average annual maximum temperature ranges from 8.7 to 17.6 °C while the
average annual minimum temperature ranges from -2.3 to 6.2 °C (PRISM Climate Group, 2012). The sub-basin has a



Mediterranean climate with cool, wet winters and dry, warm summers (Kottek et al., 2006; Snyder et al., 2002). The sub-

basin receives approximately 2,200 mm of annual precipitation, but spatially this can range from ~1,000 to 3,500 mm, primarily due to orographic effects (PRISM Climate Group, 2012). At lower elevations, the precipitation falls almost exclusively as rain, but above ~1,200 m the precipitation is generally snowfall dominated.

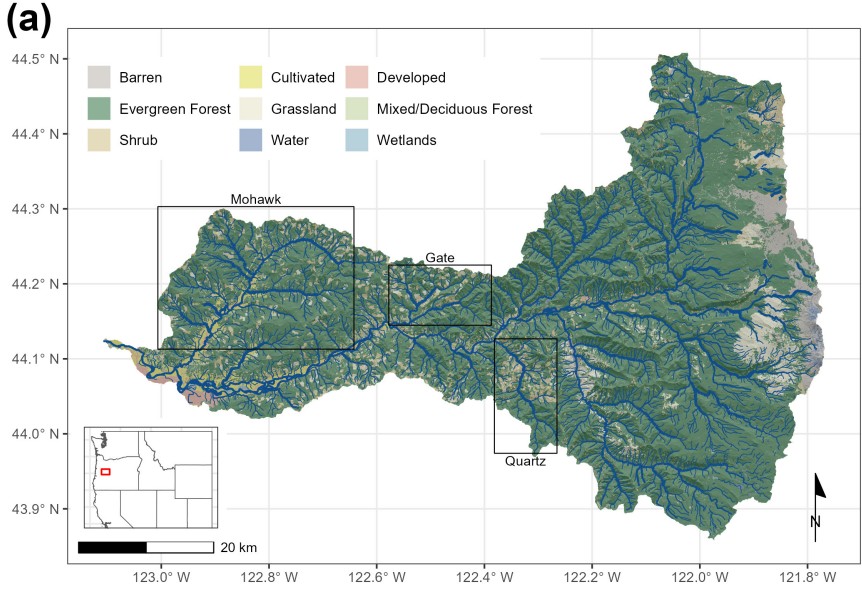

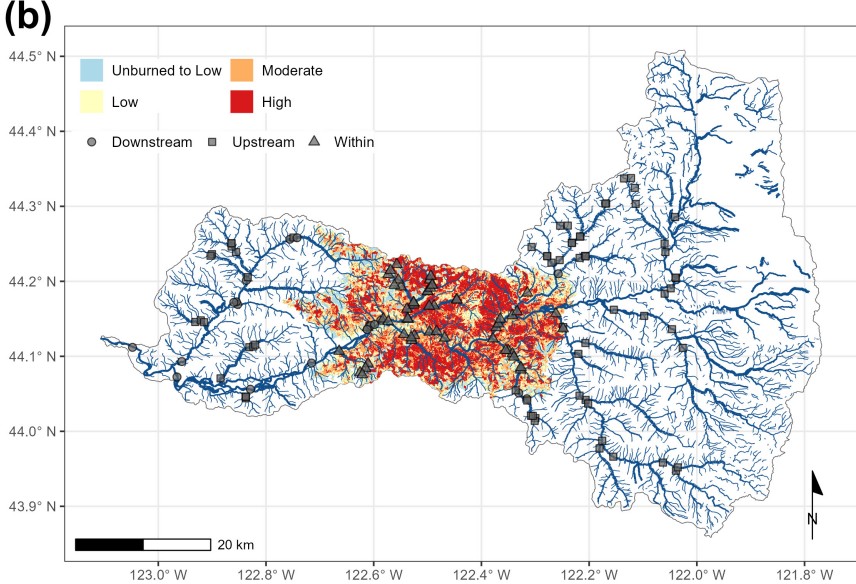

**Figure 1: (a)** The McKenzie sub-basin, Oregon USA and associated land uses. Sub-basins of particular interest are labeled. **(b)** The burn

severity of the 2020 Holiday Farm wildfire and sampling sites distributed across the stream network in the McKenzie sub-basin. The shapes indicate the location of each site relative to the Holiday Farm wildfire.



The geology of the region consists of young volcanics (Late High Cascade Volcanics) in the upper basin, which is characterized by relatively flat slopes and high permeability layers resulting substantial groundwater and springs across the region (Tague et al., 2008). In the lower portions of the sub-basin, old volcanics (Little Butte Volcanics and Late Western Cascade Volcanics) dominate and are characterized by steeper slopes with less groundwater discharge to streams (Tague et al., 2008).

In September 2020, the Holiday Farm fire burned ~18 % (629 km$^2$) of the McKenzie River sub-basin. This fire was noteworthy partially due to its size, as well as its location, which was directly on the mainstem of the McKenzie River in the middle elevations of the basin (Fig. 1b). The fire was also relatively high severity, with the area burned classified as: <1 % increased greenness, 9.2 % unburned to low, 26.9 % low, 29.4 % moderate, and 33.3 % high severity (MTBS Project, 2021).

## 2.2 Site Selection and Sample Collection

To measure the spatial variability of DOC across seasons, we selected 129 stream sites across the McKenzie sub-basin. There were 65 sites upstream of the Holiday Farm fire, 54 sites within the burn perimeter, and 10 sites downstream of the fire (Fig. 1b). We selected sites near confluences where we could collect water samples both upstream and downstream of a tributary. We also targeted an even spatial distribution of sample sites, above, within, and downstream of the Holiday Farm fire burn perimeter to encompass the landscape and climate variability within the sub-basin.

We sampled our sites four times throughout the year to capture a general seasonal variation in DOC associated with catchment wetness (Fig. 2, Table A1). The first sampling campaign was on 01 November 2022 during one of the first few rain events in the fall as the basin was starting to re-wet. Our second sampling was on 13 March 2023, during a storm in the wet season (e.g., winter). The third was on 11 June 2023, during the drying period (e.g., late spring). Lastly, we sampled on 11 September 2023, which was toward the end of the dry period (e.g., summer). The number of sites sampled varied slightly by sampling campaign due to limitations on access caused by high flows, snow, and additional wildfire restrictions (Table A1).



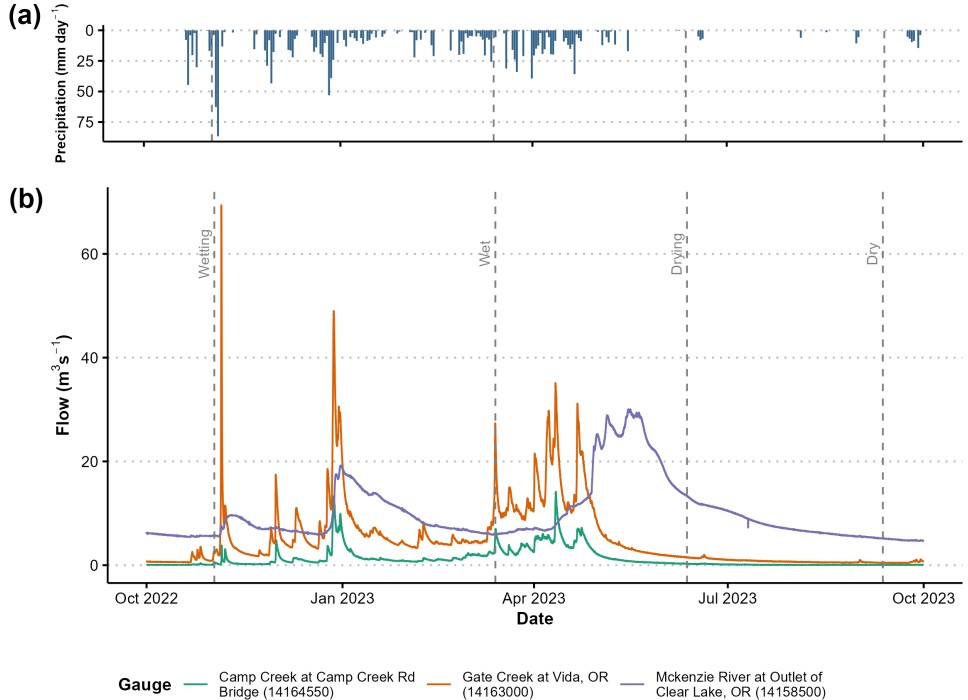

**Figure 2:** (**a**) Daily precipitation for the 2023 water year in the McKenzie sub-basin (44.2119, -122.2559) (Daly, 2023). (**b**) Discharge at three USGS stream gauges throughout the McKenzie sub-basin for the 2023 water year showing discharge patterns for the lower (Camp Creek), middle (Gate Creek) and upper (McKenzie River at Outlet of Clear Lake) parts of the sub-basin, data obtained using the dataRetrieval package in R (Cicco et al., 2018). Dates of synoptic sampling are labeled with dashed lines.

We collected stream water samples from highly mixed sections of the stream using extendable grab samplers. The grab sampler bottle was rinsed three times in the stream before collecting the sample. When feasible, samples were syringe filtered in the field using 0.2 µm PES syringe filters into acid washed and triple rinsed amber HDPE bottles. Samples that were too turbid to field filter were vacuum filtered in the lab within 24 hours of collection using 0.2 µm PES filters. Samples were kept at 4 °C after collection.

In the laboratory, we analyzed the samples for dissolved organic carbon (DOC) concentration (Shimadzu TOC-VCSH Combustion Analyzer). The analyzer acidified the samples and purged them to remove inorganic carbon, then samples were combusted at 680 °C to convert all the remaining carbon (i.e., non-purgeable organic carbon) to $CO_2$, which was measured with an infrared detector. Samples that were not able to be run within seven days of collection were frozen at -15 °C until analysis. Studies have shown that when DOC concentration are low, as was observed across our study sub-basin (Carpenter et al., 2022; Kraus et al., 2010), freezing doesn't have demonstrable impacts on DOC concentration measurements (Fellman et al., 2008; Tupas et al., 1994).



## 2.3 Spatial Stream Network Models

We used spatial stream network (SSN) models to explore the spatial variability of DOC across the stream network and determine factors influencing DOC concentrations. SSN models are multiple linear regression models with added variance to account for spatial autocorrelation along the network (Ver Hoef and Peterson, 2010). Once a model is built, the response variable can be predicted at high resolution across the network using generated prediction points. For predictions across the basin, we used prediction points spaced 1 km apart on the network. For two streams within the burn perimeter, Gate and Quartz creek, we also generated prediction points at a 100 m resolution. These streams were chosen because we had a high enough resolution of sampling points along those stream networks to generate satisfactory predictions. Models were built using the STARS toolbox in ArcMap (Peterson and Ver Hoef, 2014) and SSN package in R (Ver Hoef et al., 2014). We built both a "mean" model, including all the samples across sampling campaigns to examine the overall trends, and separate models for each sampling campaign because we expected controls on DOC concentrations to differ seasonally.

We used 13 potential explanatory variables to describe the landscape, climate, and fire at each sampling location (Table 1). We used USGS StreamStats (U.S. Geological Survey, 2019) to delineate the upstream area for each site. These polygons were used to determine the mean value of each geospatial explanatory variable at the sampling locations using zonal statistics in R (R Core Team, 2020, Table 1). Due to the number of points, basin characteristics at the prediction points were determined using the "Watershed Attributes" function in STARS; this function is less spatially explicit than StreamStats basin delineation and thus has more error which is why it was not used for the sampling locations (Peterson and Ver Hoef, 2014).

Model selection was performed using a double selection procedure (Belloni et al., 2014; Fan and Li, 2012) using linear models to determine the set of explanatory variables to use in the SSN models. For the first selection step, a linear model was fitted with DOC concentration as the dependent variable, and all the potential explanatory variables besides the variable of interest for this study, which was burn severity. The second selection step used burn severity as the dependent variable and all the potential explanatory variables. Variables were included in the final model if the variable had a *p*-value less than or equal to 0.1 in either selection step. This model selection step was performed for the mean model, which included all the observations. The selected variables from this step were also used in each of the seasonal models.

Following explanatory variable selection, the selected variables were used to create the SSN models. We fit the SSN models by first determining the best autocorrelation structure, this variance is modeled with a moving average function. Autocorrelation can be modeled using a tail-up model which only considers autocorrelation between flow connected streams, weighted by watershed area. Additionally, a tail-down model can be used which considers autocorrelation between both flow connected and unconnected streams.



**Table 1:** Descriptions of the explanatory variables used in the spatial stream network (SSN) models to model dissolved organic carbon concentrations across the McKenzie sub-basin.

| Variable | Description | Unit | Range | Median | Resolution | Source |
|---|---|---|---|---|---|---|
| Aridity | Average aridity index (P/ET) across the basin area. | – | 1.11–1.82 | 1.47 | 1000 m | (Trabucco and Zomer, 2019) |
| Available Water Capacity | Depth averaged average amount (cm/cm) of water the soils across the basin area can store that is available for plants. | cm cm$^{-1}$ | 0.10–0.23 | 0.14 | 30 m | (USDA NRCS, 2016) |
| Baseflow Index | The average percentage of streamflow attributed to groundwater discharge across the basin area. | % | 40.42–81.11 | 60.24 | 1000 m | (U.S. Geological Survey, 2003) |
| Basin Area | The area upstream from the sampling location. Basin area was determined using basin delineation in USGS Streamstats for our sampling sites. This was log transformed to reduce skewness. | ha | 0.2–3453.8 | 28.2 | – | (U.S. Geological Survey, 2019) |
| Burn Severity | The average dNBR across the basin area from the 2020 Holiday Farm Wildfire. | – | 0–773 | 0 | 30 m | (MTBS Project, 2021) |
| Elevation | The average elevation of the basin area in meters. | m | 402–1604 | 958 | 30 m | (U.S. Geological Survey, 2000) |
| Forest | Percent of the basin area classified as deciduous, evergreen, or mixed forest. | % | 48.9–100.0 | 86.0 | 30 m | (Dewitz, 2021) |
| Precipitation | Average annual precipitation (mm) across the basin area, based on 30-year normals. | mm | 1481–2832 | 2271 | 800 m | (PRISM Climate Group, 2012) |
| Sample Time | The time of day (hours) when the sample was collected from the stream. This was used to account for differences in DOC due to changes in flow throughout the day during sampling. | hour | 5.55–17.50 | 11.44 | – | – |
| Season | Seasonal basin wetness conditions describing the general antecedent conditions of the basin. | – | Wetting, Wet, Drying, Dry | – | – | – |
| Soil Clay Content | Depth averaged weight percentage of clay particles in soil. | % | 11.2–42.3 | 22.1 | 30 m | (USDA NRCS, 2016) |
| Soil Organic Matter | Depth averaged weight percentage of decomposed plant and animal residue in the soil. | % | 2.29–8.04 | 4.58 | 30 m | (USDA NRCS, 2016) |
| Soil pH | Depth averaged pH of the soil determined using the 1:1 soil-water ratio method. | – | 4.98–5.81 | 5.33 | 30 m | (USDA NRCS, 2016) |
| Topographic Wetness Index | A function of contributing area and slope describing the topographic controls on wetness. | – | 5.37–7.05 | 6.19 | 30 m | (U.S. Geological Survey, 2000) |



Finally, a Euclidian distance model can be used which disregards the stream network distances and uses 2D distance from each other across the landscape. We tested all combinations of the three autocorrelation models with their averaging functions and chose based on AIC weight and root mean squared error (RMSE). We determined the best structure using the mean model and kept the same structure for each seasonal model. Since we had repeated observations in the mean model, we added a random factor for the sampling sites. Season was also included in the mean model as a fixed effect but was not

included in the seasonal models since they were already separated by season. Lastly, the residuals of the models were checked for model issues such as lack of linearity, normality, and homoscedasticity.

### 2.4 Statistical Analysis

Statistical analysis was performed in R (version 4.2.1, R Core Team, 2020). For descriptive statistics we chose to bin the continuous burn severity dNBR values into "unburned", "low", "moderate", or "high" based on the dNBR thresholds

monitoring trends in burn severity used in classifying the burn severity for the Holiday Farm Fire (MTBS Project, 2021). dNBR values less than 40 were considered unburned, values between 41 and 320 were low severity, values between 321 and 660 were moderate severity, and values greater than 661 were high severity. To avoid extrapolating, we used an upper limit of 772 for high severity as this was the highest average dNBR at any of our sites. These dNBR thresholds were also used to estimate the impact of burn severity on DOC concentrations using the fitted dNBR coefficient in each SSN model. The

standard error of the dNBR coefficient values were used to estimate the upper and lower 95 % confidence intervals for the change in DOC for each severity group.

To test for differences in DOC concentrations across burn severity and seasonal wetness groups, we used a Gaussian family generalized liner mixed model from the glmmTMB package in R (Brooks et al., 2017), which allowed us to examine both

the mean and variance across severity and wetness groups while accounting for the heteroscedasticity between groups and repeated measures across sites. Mean DOC was modeled using burn severity, seasonal wetness, and the interaction between the two, with site as a random variable. The variance was modeled using burn severity and seasonal wetness without an interaction.

To determine the relative importance of the explanatory variables in each SSN model the values of each explanatory variable were standardized to a mean of 0 and a standard deviation of 1 by subtracting the mean value and dividing by the standard deviation of each variable. This sets all coefficients on the same scale, so larger coefficients indicate a larger impact on DOC concentrations while the sign indicates the direction of the impact (positive vs negative).



# 3 Results

## 3.1 Spatial Trends in Dissolved Organic Carbon Concentrations


We used our observed measurements of dissolved organic carbon (DOC) to predict DOC concentrations across the stream network using spatial stream network (SSN) models; those predictions were used to visualize spatial patterns across the sub-basin. Overall, we observed a general spatial trend of low DOC concentrations in the eastern headwaters of the sub-basin, with concentrations increasing as flow moved downstream (West, Fig. 3). This trend was particularly visible in the wetting

and dry seasons, but also present in the wet and drying seasons. Contrary to expectations, we did not observe an obvious fire signal as flow moved through the burned area. The measured and predicted DOC concentrations within the fire perimeter did not appear to be distinctively lower or higher than the surrounding areas and were consistent with the East to West pattern observed. We also noted that the mainstem remained low in DOC throughout all the seasonal wetness conditions, regardless of higher DOC inputs from tributaries, especially in the wetting season. One of those tributaries that contributed high DOC

concentrations across seasons was the Mohawk tributary on the NW portion of the sub-basin (Fig. 1). It consistently had some of the highest concentrations during each sampling period. The Mohawk tributary is primarily dominated by evergreen forest (80.2 %) and also has a relatively low baseflow index (47.4 %).



**Figure 3:** Maps showing the observed (left) and predicted (right) dissolved organic carbon concentrations (DOC) across the McKenzie sub-basin, OR for the four seasonal wetness conditions. The wildfire perimeter is shown in gray. Predictions were obtained using the final fitted spatial stream network (SSN) models and prediction points spaced every 1km along the stream network. The size of the points in the predictions indicates the prediction error associated with the point.



## 3.2 Trends in DOC Associated with Burn Severity and Seasonal Wetness Conditions

To further explore the impact of burn severity on DOC in the basin, we summarized the observed DOC data by burn severity

groups to explore the impact of burn severity numerically. Overall, DOC concentrations were relatively low for all severity groups. The mean for the unburned sites was $1.27 \pm 1.12$ mg $L^{-1}$ which was slightly higher than the low severity sites with a mean of $1.16 \pm 0.83$ mg $L^{-1}$. The moderate severity sites had a mean concentration of $1.22 \pm 0.58$ mg $L^{-1}$. Lastly the high severity sites had a mean concentration of $1.08 \pm 0.46$ mg $L^{-1}$. A generalized liner mixed model (GLMM) indicated that there was weak to minimal evidence of differences across severity groups (*p*-values: 0.0984–0.7867). While there were no major

shifts in DOC concentration, the model did indicate significant differences in variance between severity groups. Specifically, the variance of the low severity group was 0.36 times (95 % CI: 0.33–0.39) the variance of the unburned group. The variance for the moderate severity group was 0.13 times (95 % CI: 0.12–0.15) the unburned variance and the high severity group was 0.14 times (95 % CI: 0.10–0.19) the unburned variance. While the moderate and high severity groups had smaller variance than the low and unburned groups, there was not a distinguishable difference in variance between the moderate and high

severity groups. The model estimated the ratio of variance between the moderate and high severity groups to be 1.04 (95 % CI: 0.76–1.44), but the 95 % CI crossed 1, indicating that at an α level of 0.05, the variance between the two groups is indistinguishable.

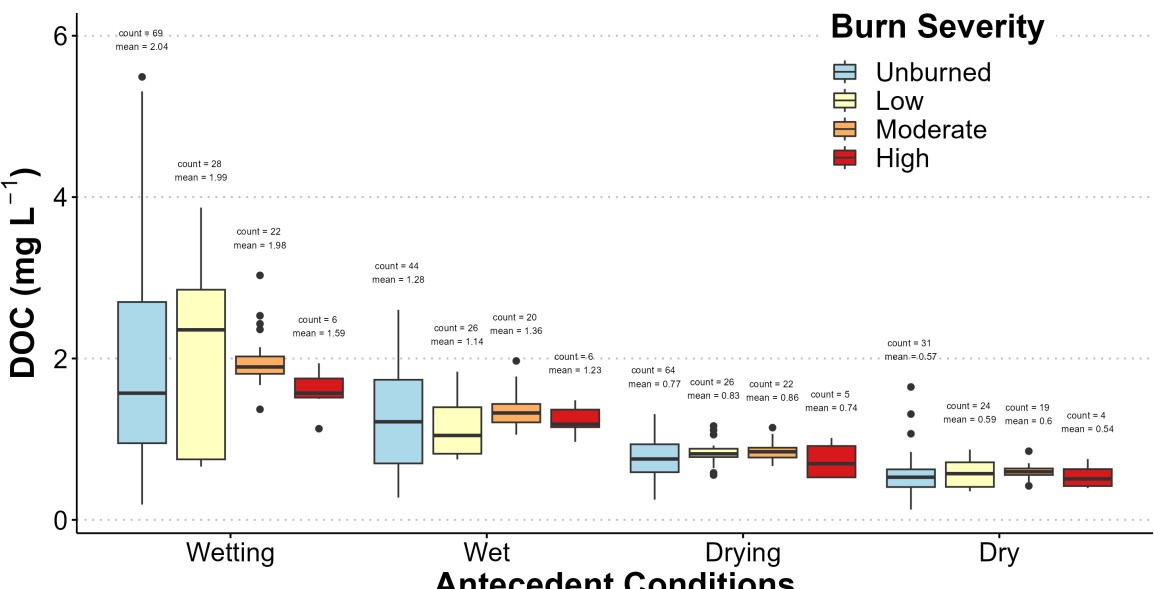

**Figure 4:** The distribution of in-stream DOC concentrations in the McKenzie sub-basin, OR for each burn severity group over the four sampling campaigns (01 September 2022, 13 March 2023, 11 June 2023, 11 September 2023), labeled by the seasonal wetness conditions. The number of samples and mean are labeled above each group.



In addition to burn severity effects, we also noted that antecedent seasonal wetness conditions had an impact on DOC concentrations. Unlike burn severity, our GLMM model indicated that there was strong evidence ($p < .001$, Table A2) of differences in mean DOC concentrations different across seasonal wetness conditions. DOC concentrations were highest in the wetting season, followed by the wet, drying and dry seasons (Fig. 4). Additionally, the variance of DOC concentrations was seasonally dependent, with the greatest variance during the wetting season, with a variance that was 6.7-times (95 % CI:

6.19–7.33) greater than the variance during the wet season. Similarly, the wet season had a variance that was 7.5-times (95 % CI: 6.26–9.06) greater than during the drying season. Statistically, there was no evidence for a difference in the variance between the drying season and the dry season (ratio= 0.90, 95 % CI: 0.66–1.23).

### 3.3 Spatial Stream Network Models of DOC

We created a spatial stream network (SSN) model using the DOC data collected in our sampling campaigns to predict the

landscape, climate and wildfire factors influencing DOC concentrations and estimate the impact of burn severity across the McKenzie sub-basin, OR. Post-double-selection was used to select 12 of the 14 potential explanatory variables for inclusion in the final model (Table 2). Specifically, annual precipitation and topographic wetness index did not have sufficient empirical support to support inclusion. A search across all potential spatial autocorrelation structures and models resulted in the selection of exponential tail-up and gaussian Euclidean models (Table 3). Overall, the model fit was moderately strong,

with a leave one out cross validation $R^2$ of 0.602 (Table 3). Despite this, the mean model had a large nugget (38.0 %), suggesting there is quite a bit of variance unaccounted for in the model. We standardized our non-categorical explanatory variables, allowing us to compare model coefficients. Overall, the most important factors influencing DOC across the sub-basin was baseflow area, this was followed by burn severity, aridity index, and soil pH. All four exhibited negative relationships with DOC concentrations (Fig. 5). Conversely, an increase in the percentage of soil organic matter was associated with increased

DOC concentrations (Fig. 5). Other factors that were less important but still positively related to DOC were soil clay percentage and available water capacity, while sample time, log of basin area, and percent forested all had negative relationships with DOC (Fig. 5). As expected, given the variation in seasonal concentrations, antecedent conditions also had a large impact on DOC concentrations. The wetting season had the highest mean DOC concentrations, with predicted decreases in mean DOC of 0.85 mg L$^{-1}$ in the wet season, 1.18 mg L$^{-1}$ the drying season, and 1.50 mg L$^{-1}$ in

the dry season.




**Table 2:** Standardized coefficients for the covariates used in the mean model and the four seasonal spatial stream network (SSN) models describing the concentration of dissolved organic carbon (DOC) within the McKenzie sub-basin, OR. All non-categorical explanatory variables were standardized by calculating the z-score for each individual value in the explanatory variable.

| | Season | | | | |
|---|---|---|---|---|---|
| **Covariates** | **Mean** | **Wetting** | **Wet** | **Drying** | **Dry** |
| Aridity Index | -0.140 | -0.149 | -0.117 | -0.100 | -0.143 |
| Available Water Capacity | 0.006 | -0.187 | -0.014 | -0.034 | 0.031 |
| Baseflow Index | -0.354 | -0.562 | -0.543 | -0.141 | -0.248 |
| Soil Clay Percentage | 0.029 | 0.250 | -0.178 | 0.008 | 0.028 |
| Burn Severity (dNBR) | -0.204 | -0.502 | -0.049 | -0.033 | 0.016 |
| Elevation | -0.058 | 0.021 | 0.043 | 0.071 | 0.002 |
| Percent Forested | -0.037 | -0.012 | 0.005 | -0.029 | -0.022 |
| log of Basin Area | -0.050 | -0.214 | -0.097 | -0.016 | 0.042 |
| Soil pH | -0.104 | -0.006 | -0.119 | 0.012 | 0.030 |
| Sample Time | -0.053 | 0.074 | 0.126 | 0.002 | 0.006 |
| Soil Organic Matter | 0.095 | 0.507 | -0.023 | 0.077 | 0.001 |
| Intercept | 2.037 | 2.030 | 1.239 | 0.813 | 0.605 |

**Table 3:** The proportion of the variance explained by the covariates, the autocorrelation functions (tail-up and Euclidean), and the unexplained variance (nugget). The total autocorrelation variance is the sum explained by the tail-up and Euclidean components (shown in italics). Below the variance metrics is the $R^2$ determined from the leave one out cross validation and the root mean squared error for each of the mean model and the four seasonal spatial stream network (SSN) models of dissolved organic carbon (DOC) concentrations in the McKenzie sub-basin, OR.

| | Season | | | | |
|---|---|---|---|---|---|
| | **Mean** | **Wetting** | **Wet** | **Drying** | **Dry** |
| Covariate $R^2$ | 0.602 | 0.405 | 0.544 | 0.135 | 0.403 |
| Total Autocorrelation | 0.018 | 0.595 | 0.435 | 0.635 | 0.522 |
| *tail-up* | *<0.001* | *0.174* | *0.314* | *0.253* | *0.510* |
| *Euclidean* | *0.018* | *0.421* | *0.12* | *0.382* | *0.011* |
| Nugget | 0.38 | <0.001 | 0.002 | 0.083 | 0.034 |
| Cross Validation $R^2$ | 0.606 | 0.879 | 0.823 | 0.456 | 0.436 |
| Root Mean Squared Error | 0.586 | 0.448 | 0.208 | 0.165 | 0.178 |



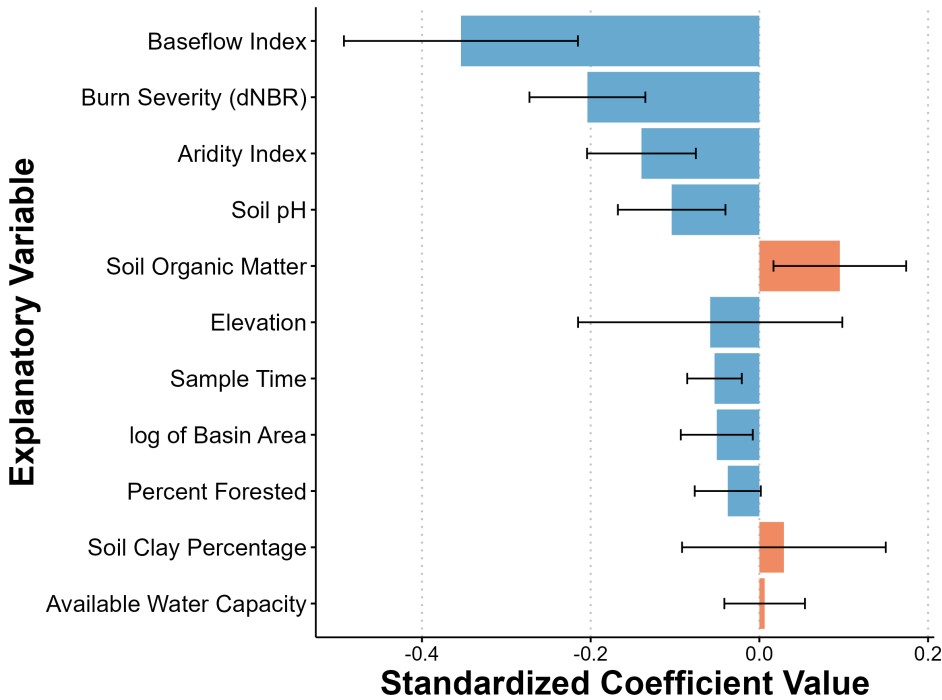

**Figure 5:** The standardized coefficient values of explanatory variables driving spatial differences in dissolved organic carbon (DOC) concentrations across the McKenzie sub-basin, OR from 2022 to 2023, two years post-fire. Values were determined from standardizing the explanatory variables prior to developing a spatial stream network (SSN) model using water quality samples across four distinct seasonal antecedent wetness conditions. The color indicates the direction of the relationship between the variable and DOC concentrations. Error bars indicate the standard error from only the final model. They do not represent the error associated with model selection performed to determine the final model.

Since seasonal antecedent wetness conditions were strongly related to DOC concentrations and the mean model exhibited substantial unexplained variation, we suspected that the impact of some of the other factors influencing DOC concentration may change seasonally. Because of this, we created a unique SSN model for each antecedent condition. The wetting and wet models exhibited improved fits with leave one out cross validations $R^2$ of 0.879 for the wetting period and 0.823 for the wet period (Table 3). The drying and dry models had poorer fits, with cross validation $R^2$ values of 0.456 for the drying period and 0.436 for the dry period. Compared to the mean model, all models had smaller nuggets, suggesting the fixed effects and spatial autocorrelation did a better job of explaining the DOC concentrations in the separate seasonal models (Table 3). Notably, while ~50 % of the variation in DOC was explained by the covariates in the wetting, wet, and dry season models, they only accounted for 13.5 % of the variance in the drying model (Table 3). Interestingly, the majority of spatial autocorrelation was accounted for by the tail-up (flow connected) structure in the wet and dry seasons while the Euclidian structure was most important in the wetting and drying seasons (Table 3).





In terms of factors related to DOC concentrations, the importance and direction of many covariates changed seasonally (Fig. 6). Notably, the percentage of clay in the soil influenced DOC concentrations in both the wetting and wet seasons. However, it was associated with increased DOC during the wetting season but decreased DOC concentrations during the wet season. The percentage of soil organic matter also influenced DOC concentrations in the wetting and drying seasons but appeared to have minimal importance in the wet and dry seasons. Despite some factors being seasonally variable, the most influential

factor in the mean model was the baseflow index, which remained the most important variable in all seasons. As the baseflow index increased in a sub-catchment, DOC concentrations consistently decreased. Similarly, as the aridity index increased, DOC concentrations consistently decreased. During the drying and dry seasons the aridity index was the second most influential variable on DOC concentrations. However, the aridity index tended to have less influence on DOC during the wetting and wet seasons.


Contrary to expectations, burn severity was not the most influential factor on DOC concentrations in any season (Fig. 6). Across seasonal wetness conditions, it was most important in the wetting season where it was one of the top three factors, similar in magnitude to the impact of soil organic matter. However, in the other seasons it was only of moderate (wet and drying) or minimal importance (dry). Overall, like the mean model, the models predicted that an increase in burn severity

would decrease DOC concentrations, despite the dry season model prediction of a slight increase in DOC during that season.

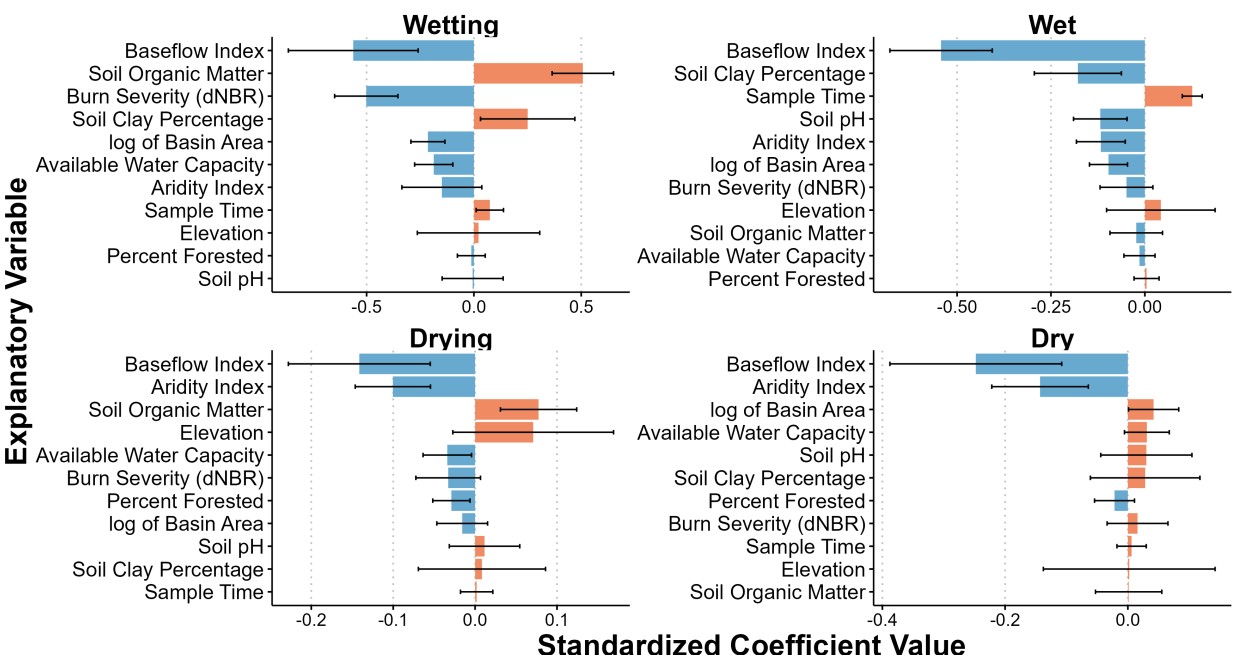

**Figure 6:** The standardized coefficient values of explanatory variables driving spatial differences in dissolved organic carbon (DOC) concentrations across the McKenzie sub-basin, OR during four distinct seasonal antecedent wetness conditions. Values were determined from standardizing the explanatory variables prior to developing spatial stream network (SSN) models. The color indicates the direction of
the relationship between the variable and DOC concentrations. Error bars indicate the standard error from only the final model. They do not represent the error associated with model selection performed to determine the final model.



To better contextualize our results, we used the burn severity thresholds determined by MTBS for the Holiday Farm fire to estimate the average change in DOC for each burn severity group from our SSN models (Fig. 7). For the high severity burn

classification, DOC concentrations were estimated to decrease between -1.40 to -1.64 mg L$^{-1}$ during the wetting season. The impact was much lower in the other seasons, with an estimated decrease of -0.14 to -0.16 mg L$^{-1}$ for the wet season and -0.09 to -0.10 in the drying season. In the dry season, DOC was predicted to increase between 0.03 to 0.04 mg L$^{-1}$. However, it should be noted that the wetting season was the only season where the 95 % confidence interval did not cross 0, indicating some uncertainty in the directionality of burn severity influence on DOC concentrations during the wet, drying, and dry

seasons (Fig. 7).

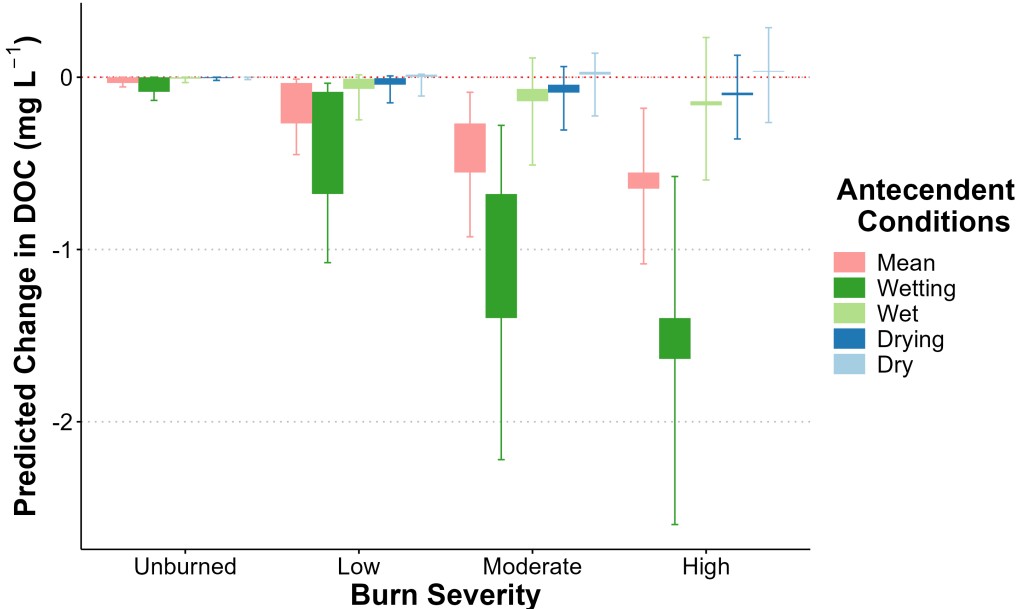

**Figure 7:** The range of predicted change in stream dissolved organic carbon (DOC) for the mean change and each seasonal antecedent condition at each binned burn severity level for the McKenzie sub-basin, OR. Predictions are based on the burn severity coefficient in the fitted spatial stream network (SSN) models. Burn severity groups are based on the dNBR thresholds determined by monitoring trends in

burn severity (MTBS) for the 2020 Holiday Farm wildfire. The bar shows the range in changed based on the low and high threshold values for each burn severity group. The error bars are based on the standard error of the burn severity coefficient, where the upper bar is the upper 95 % confidence interval for the low threshold values and the lower bar is the lower 95 % confidence interval for the high threshold values.

## 4 Discussion

**4.1 Landscape factors were more influential than burn severity in controlling DOC concentrations.**

In our analysis of 129 sites across the stream network of the McKenzie River sub-basin in Oregon, USA we did not observe an obvious effect from the Holiday Farm wildfire on the spatial pattern of DOC (Fig. 3). Wildfires are known to impact organic matter availability on burned hillslopes and shift hydrologic flow paths (Atwood et al., 2023; Jung et al., 2009; Onda



et al., 2008; Rey et al., 2023; Stoof et al., 2014; Certini et al., 2011), which theoretically influence the delivery of DOC to

streams. Therefore, we expected DOC concentrations would be influenced by the fire substantially, as streamflow moved from upstream to within the burn perimeter and to downstream sites. However, there was not a clearly evident impact of the wildfire on DOC concentrations across the sub-basin. Interestingly, Rüegg et al. (2015) observed a similar lack of response of DOC concentrations to burning in a prairie basin in Kansas, USA. They posited that this was due to limited changes in terrestrial carbon due to the generally low severity of the controlled burns at their study sites. However, this was likely not

the case in our basin due to the mixed severity burn of the Holiday Farm fire, including ~63 % of the area burning at moderate to high burn severity. Moreover, research in high burn severity sites across the Holiday Farm fire area, found evidence of decreases in the percentage of soil organic carbon in the first 2 cm of soil (56 % change) and in the particulate organic carbon fraction (61 % change, Katz et al., 2023). The lack of a fire signal on the DOC concentrations may be related to the thick and active organic horizons and extremely high rates of saturated hydraulic conductivity in soils of the Pacific

Northwest (Jarecke et al., 2021), which would have enabled infiltration and vertical percolation of water and DOC through the soil profile despite the burn. A similar phenomenon was observed in NE Victoria Australia as soil saturated hydraulic conductivity remained very high (100–1,000 mm h$^{-1}$) following wildfire (Sheridan et al., 2007). Additionally, landscape characteristics such as groundwater contributions, elevation, land use, slope, aspect, and aridity may also have influenced DOC concentrations across space and muted the effects from the wildfire. A similar effect was observed in the Northwest

Territories, CAN—site factors explained ~50 % of the variation in DOC concentrations compared to only ~5 % explained by wildfire following repeat sampling across 50 burned and unburned sites (Hutchins et al., 2023).

Despite the lack of a clear effect of the wildfire on the spatial pattern of DOC, we did observe a pattern of low DOC concentrations in the high elevation headwaters, with DOC generally increasing at lower elevation sites near the outlet of the

basin (Fig. 3). This observed pattern could be attributable to a number of landscape factors. First, elevation is tightly coupled with air temperature in our basin, with strong lapse rates driving warmer temperatures at lower elevations. Warmer temperatures can increase rates of litter decomposition (Salinas et al., 2011), which could have contributed to larger DOC source pools in soils and greater stream DOC concentrations at lower elevations. A similar trend was observed in Germany where there was strong evidence of increased DOC concentrations with decreasing elevation and increasing air temperatures

(Borken et al., 2011). Similarly, DOC was predicted to increase by 0.71 mg L$^{-1}$ with a 500 m decrease in elevation in the Rocky Mountains of Colorado, USA (Rodríguez-Jeangros et al., 2018). However, the spatial pattern of DOC at our sites could also be due to the correlation of elevation with other landscape factors such as land use. Rodríguez-Jeangros et al. (2018) also noted a greater proportion of agricultural and urban land use areas at lower elevations, which was correlated with increasing concentrations of DOC. Similarly, in our basin, agriculture land use is increased in the lower one-third of the

basin, centered around the mainstem and Mohawk tributary (Fig. 1). As agricultural areas are often sources of high organic matter (Chen et al., 2021; Shang et al., 2018) the spatial pattern of land uses may have influenced the spatial DOC concentrations across the McKenzie River sub-basin. Indeed, some of the highest DOC concentrations were observed in the



Mohawk tributary at the lower end of the basin (Fig. 3). While spatial patterns of DOC have also been tied to locations within watersheds with substantial wetland or peatland influence (Ågren et al., 2007; Dawson et al., 2001; Dupas et al.,
2021; Piatek et al., 2009; Vidon et al., 2014), this was unlikely a substantial influence in our study as the McKenzie sub-basin has minimal wetland influence.

Increases in DOC concentrations from headwaters to outlet may have been influenced by the relative proportion of groundwater inputs along the stream network. The headwaters of the McKenzie sub-basin are comprised of High Cascades
geology, which is known for high proportions of deep groundwater inputs (Jefferson et al., 2006). Thus, we were not surprised that our SSN models suggested that groundwater inputs, as represented by the baseflow index (Fig. 5 & 6), were strongly influential on DOC concentrations across the McKenzie sub-basin. Indeed, the variability in DOC due to differences in groundwater inputs may have been greater than the variability created by the Holiday Farm wildfire, muting the impacts of burn severity on DOC. This is consistent with previous work in the McKenzie sub-basin, which found the lowest DOC
concentrations to be associated with areas with significant groundwater discharge (Kraus et al., 2010). Others have also noted the importance of typically low DOC groundwater inputs in controlling the spatial patterns of DOC. For example, for two basins in Northeast US, groundwater seeps led to headwater streams with some of the lowest DOC concentrations observed across the stream network (Vidon et al., 2014; Zimmer et al., 2013). Additionally, spatial studies in Sweden and Japan measured significant negative relationships between the percentage of groundwater and DOC concentrations (Egusa et
al., 2021; Peralta-Tapia et al., 2015). The importance of groundwater on DOC concentrations after wildfire was also noted following in Alberta, CAN, where there was no measurable impact of wildfire on DOC concentrations in fens (Davidson et al., 2019) or boreal lakes (Olefeldt et al., 2013). Both attributed this lack of wildfire effect to the regulation of DOC concentrations due to the processes of selective adsorption, degradation, and desorption as water and DOC moves slowly downwards through mineral soils to groundwater tables (Kaiser and Kalbitz, 2012). This is consistent with studies from
California and Alaska, USA where the role of hydrology and hydrologic connectivity of burned hillslopes and streams were also observed to regulate DOC concentrations (Barton et al., 2023); (Larouche et al., 2015). The importance of landscape variables over fire impacts highlights the importance of including local landscape characteristics in models to predict post-fire DOC responses or to enable interpretation of empirical DOC data.

Despite variable DOC inputs from tributaries, the mainstem remained remarkably stable with low DOC concentrations along its length (Fig. 3). This pattern of homogenization has been noted by others (Bhattacharya and Osburn, 2020; Creed et al., 2015) who attributed the lower variability in DOC concentrations at high stream orders to hydrological averaging and a dominance of in-stream processes. Our findings suggest that the impacts from large wildfires, which affect a large portion of the stream network may also be "averaged out". While additional research is needed to better quantify how wildfire impacts
on DOC concentrations propagate through a stream network, future post-fire studies should carefully consider the scale at which measurements are collected to account for the impact of potentially confounding landscape factors and hydrologic



averaging along the stream network. Future work should also consider the downstream extent to which wildfires impact the stream network water quality, as effects on stream network biogeochemistry have been observed (Ball et al., 2021).

## 4.2 Wildfire decreased variability of DOC concentrations and led to seasonal variable decreases in concentration

We observed no differences between the mean DOC concentrations across burn severity groups, however we observed that the variability in DOC concentrations was lowest in the moderate and high severity burned areas (Fig. 4). This change in variability has been relatively unexplored in terms of wildfire effects, as researchers typically focus on the magnitude of change in DOC following fire. However, understanding the variability of post-fire responses is critical in making predictions of post-fire water quality impacts. We believe that the decrease in variability at the higher severities is caused by

homogenization of the landscape in the wildfire, removing factors (i.e. soil characteristics, vegetation) that would normally lead to spatial variability in DOC. This is similar to what was found thirteen years post-fire in Colorado (Chow et al., 2019). Basins that burned at high extent (>75 % of area burned, 50–60 % at high severity) exhibited less seasonal variability than those burned at moderate extent or unburned which was linked to slow vegetation recovery and bare landscapes. However, the decrease in variability may not be observed immediately post-fire. A two-year study following wildfire in California,

found no obvious differences in variability in the first year post-fire, but in the second year one of the two burned sites exhibited noticeably less variability than the unburned site (Uzun et al., 2020). However, initial flushing of post-fire material may lead to short-term increases in variability in burned areas. In the first year post-fire, DOC variability was increased following wildfire in Colorado and Utah, US likely due to debris flows and ash flushed from the burn scar (Crandall et al., 2021; Hohner et al., 2016). Despite the likely homogenization of the post-fire landscape observed here in the two years post

fire, remaining landscape characteristics remained the dominate control on the magnitude of post-fire DOC concentrations.

Burn severity was never the most important factor controlling DOC, but it was still important in predicting DOC in the wetting season (Fig. 6). We hypothesized that during the wet season, hydrologic flow paths would shift to more shallow pathways, increasing the connectivity between burned hillslopes and streams, thereby increasing the fire signal across the

network (Fellman et al., 2009). However, this was not what we observed, with a stronger fire signal during the wetting season. This could be partially due to the wet season storm we captured, which was a relatively small storm and may not have led to as much hillslope connectivity as we expected (Fig. 2). If we had sampled our sites during a larger storm, we may have found burn severity was more important. Regardless, the variability in the importance of burn severity across antecedent conditions is a critical finding, as it could help explain the variable nature of wildfire impacts on DOC observed

across the literature. A review of post-fire impacts found that of the 27 post-fire effects reviewed, the range of DOC concentration changes had the most variability between the 25th and 75th quantiles (Paul et al., 2022). Given our results, it is likely that this spread in post-fire impacts is partially due to studies sampling across different antecedent conditions. Notably, two of the studies which found that wildfire had no impact on DOC used long-term periodic sampling which likely included



many seasonal periods where burn severity was not important (i.e. drying and dry seasons), which could have contributed to
the conclusion that wildfire had no impact (Mast and Clow, 2008; Wagner et al., 2015).

Our SSN models predicted significant decreases in DOC with increasing burn severity during the wetting period, with
minimal and uncertain decreases in the other seasons (Fig. 7). There are two potential explanations for this. First, decreases
in evapotranspiration due to vegetation loss post-fire could be leading to increased groundwater to burned streams. Drops in
ET have been measured following several wildfire in different regions (Ma et al., 2020; Nolan et al., 2014; Poon and
Kinoshita, 2018). While recent work has reported that wildfire can lead to increased groundwater contributions (MacNeille
et al., 2020; Rey et al., 2023), previous studies in the basin have identified the lowest DOC concentration in the areas with
high groundwater inputs (Kraus et al., 2010). This hypothesis was also proposed by Santos et al. (2019) to explain the
decreases in DOC they measured during baseflow periods in California, USA. A second possibility is that we observed lower
DOC due to combustion loss of soil organic matter during the wildfire, which decreased sources of carbon within the basin.
This explanation matches well with our seasonal findings, where we primarily observed an effect of burn severity in the
wetting period (Fig. 6). Previous work in the region has noted that DOC is often highest during the wetting season, due to
flushing of available organics built up during the dry summer period, with fresh plant residues playing a significant role
(Sanderman et al., 2008). Indeed, a previous study within our basin found that during the rising limb of storms, DOC was
sourced from the organic horizon (van Verseveld et al., 2008). So, a decrease in soil organic matter and loss of vegetation
post-fire would likely mute this flushing behavior, resulting in decreased DOC during the wetting period. A post-fire soil
study in the McKenzie sub-basin found that total soil organic carbon was decreased along a burn severity gradient (Katz et
al., 2023). Other studies have also proposed decreased soil organic stocks as the cause of decreased DOC observed post-fire
in California and Alaska, USA (Betts and Jones, 2009; Santos et al., 2019).

**5 Conclusion**

In our study, we quantified DOC concentrations across four distinct seasonal wetness conditions at high spatial resolution
across a large sub-basin affected by wildfire in the western Cascade Mountains of Oregon. This enabled us to relate DOC to
landscape characteristics, sub-catchment burn conditions, and seasonal wetness conditions. Our findings suggest that
increased burn severity may decrease DOC concentrations in streams, most notably in the wetting season, albeit this result is
complicated by landscape hydrologic pathways and catchment characteristics.  For example, we found that DOC responses
to wildfire may be substantially dampened in systems dominated by deep, sub-surface flow paths of water and groundwater
discharge. While our results provide additional context for the wide variability of post-fire DOC responses reported in the
literature, a universal understanding of the response of DOC to wildfire remains unresolved. For example, while we observed
little shifts in DOC concentrations following wildfire, our study did not address the potential changes in dissolved organic
matter character that may occur, which can influence its fate in the environment. Shifts in DOM character to more



recalcitrant forms could create challenges for downstream drinking water treatment or impacts on aquatic ecosystems even in the absence of changes in DOC concentrations (Hohner et al., 2016), and may be directly linked to drainage area burn severity during storm events (Roebuck Jr. et al., 2022). Thus, further work exploring how DOC concentrations and DOM character changes with burn severity across both space and time could further improve our understanding of the mechanisms

of delivery of DOC from burned hillslopes to streams (Roebuck Jr. et al., 2022, 2023). Specifically, DOM character indices have been shown to be a relatively simple way to understand organic matter sources (Hood et al., 2006). Similarly, stable carbon isotopes could also be quantified following wildfire to predict the age and source of DOC fluxes and provide insights into the mechanisms of DOC movement through soils (Sanderman et al., 2008). High frequency storm sampling in burned and unburned sites, particularly in the fall, could help us better understand the mechanisms of post-fire DOC changes as

hysteresis metrics can also help determine controlling processes of DOC transport in burned basins (Liu et al., 2021). Finally, it would be valuable to repeat this high spatial density sampling experiment in different climates to help understand the generalizability of our results.




## Appendix A

**Table A1:** Descriptions of the date, start and end time of the sampling campaigns, the average rainfall intensity during the sampling period, and antecedent precipitation conditions (API 1,7, and 31), and number of sites sampled. Sampling occurred across the McKenzie sub-basin in Oregon, US. Precipitation data is based on the PRIMET station in the HJ Andrews, OR (44.2119, -122.2559, Daly, 2023).

| Season | Sampling Date | Start Time (PST) | End Time (PST) | Storm Volume (mm) | Average Intensity (mm hr$^{-1}$) | API 1 (mm) | API 7 (mm) | API 31 (mm) | Sites Sampled |
|---|---|---|---|---|---|---|---|---|---|
| Wetting | 01 November 2022 | 5:30 | 17:15 | 7.62 | 0.65 | 13.97 | 45.72 | 123.70 | 131 |
| Wet | 13 March 2023 | 7:00 | 15:15 | 16.51 | 2.00 | 9.91 | 54.36 | 202.44 | 99 |
| Drying | 11 June 2023 | 6:30 | 17:30 | 0.00 | 0.00 | 0.00 | 0.51 | 17.53 | 122 |
| Dry | 11 September 2023 | 6:45 | 17:15 | 0.00 | 0.00 | 0.00 | 0.00 | 18.03 | 81 |

**Table A2:** Comparisons of differences in mean DOC concentrations across season using a glmmTMB model and the emmeans package in R (Lenth, 2021; Brooks et al., 2017). The degrees of freedom (df), t-ratio, and p-value are reported for each contrast.

| Contrast | df | t-ratio | p-value |
|---|---|---|---|
| wetting - wet | 392 | 7.248 | <0.001 |
| wetting - drying | 392 | 12.935 | <0.001 |
| wetting - dry | 392 | 15.447 | <0.001 |
| wet - drying | 392 | 11.609 | <0.001 |
| wet - dry | 392 | 16.947 | <0.001 |
| drying - dry | 392 | 10.668 | <0.001 |



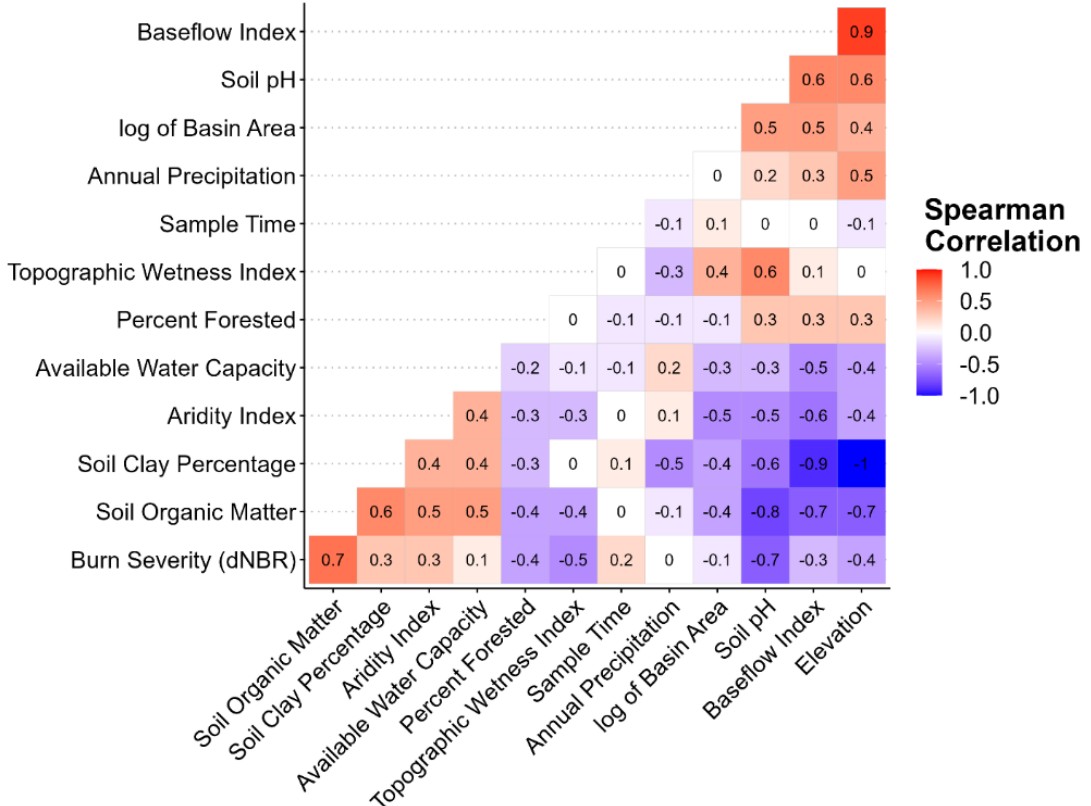


**Figure A1:** Spearman correlation matrix showing the correlation between the explanatory variables used in model selection for the spatial stream network models in the McKenzie sub-basin, OR. For descriptions of each variable see Table 1.





**Figure A2:** Leave one out cross validation results for the mean and seasonal spatial stream network (SSN) models. The red dashed line indicates a 1:1 line. Potential outliers are labeled with the site number and indicated by red points.





**Figure A3:** Dissolved organic carbon (DOC) across the McKenzie sub-basin, OR during the wetting season. (**a**) Observed DOC concentrations with the extents shown for plots b and c, (**b**) predicted DOC concentrations across the Gate Creek stream network at a 100 m resolution (**c**) predicted DOC concentration across the Quartz Creek stream network at a 100 m resolution. Predictions were made with a spatial stream network (SSN) model.



**Figure A4:** Dissolved organic carbon (DOC) across the McKenzie sub-basin, OR during the wet season. (**a**) Observed DOC concentrations with the extents shown for plots b and c, (**b**) predicted DOC concentrations across the Gate Creek stream network at a 100 m resolution (**c**) predicted DOC concentration across the Quartz Creek stream network at a 100 m resolution. Predictions were made with a spatial stream network (SSN) model.






**Figure A5.** Dissolved organic carbon (DOC) across the McKenzie sub-basin, OR during the drying season. (**a**) Observed DOC
concentrations with the extents shown for plots b and c, (**b**) predicted DOC concentrations across the Gate Creek stream network at a 100
m resolution (**c**) predicted DOC concentration across the Quartz Creek stream network at a 100 m resolution. Predictions were made with a
spatial stream network (SSN) model.






**Figure A6:** Dissolved organic carbon (DOC) across the McKenzie sub-basin, OR during the dry season. (**a**) Observed DOC concentrations with the extents shown for plots b and c, (**b**) predicted DOC concentrations across the Gate Creek stream network at a 100 m resolution (**c**) predicted DOC concentration across the Quartz Creek stream network at a 100 m resolution. Predictions were made with a spatial stream network (SSN) model.



**Code and Data availability:** The data and code that support the findings of this study are openly available at Scholars
Archive at Oregon State University, https://doi.org/10.7267/zc77sz60m.

**Author contribution**: KB and AMP were responsible for funding acquisition, supervision, and writing – review & editing.
KB provided the resources and conceptualization. KW performed the investigation, formal analysis, methodology,
visualization, and writing – original draft preparation.


**Competing interests:** The authors declare that they have no conflict of interest.

**Acknowledgements:** We would like to thank Dustin Gannon for his help with the statistics in this paper. Funding to support
collection of this data set was provided by the U.S. Forest Service agreement numbers 22-JV-11261952-071 and 23-JV-
11261954-057 and from the US Department of Energy, Office of Science, Biological and Environmental Research, as part of
the Environmental System Science (ESS) Program to the River Corridors Science Focus Area at the Pacific Northwest
National Laboratory (PNNL). PNNL is operated for DOE by Battelle Memorial Institute under contract DE-AC05-
76RL01830.

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
