# Peer review of "The influence of burn severity on dissolved organic carbon concentrations across a stream network differs based on seasonal wetness conditions"

_EGUsphere, 2024_

## Author Response (AR1)

**EDITOR COMMENTS**
Thanks for your responses to the comments that we received from the two reviewers. I agree with their comments and your responses, especially also how you plan to incorporate their suggestions in the revised manuscript. I had a couple of questions and comments in my initial review of your manuscript, which were already raised by the reviewers, so I do not need to repeat that. However, most importantly, I share the concerns about the delay of sampling post-fire and how that may have impacted your results. While I understand your response that other studies reported data obtained from samples long after fire events, I think you should address these concerns in detail. I think it is important to state how different your results may be compared to samples that were obtained much sooner and therefore likely much less altered?

**REVIEW 1**
Thank you for taking the time to provide thoughtful and helpful comments to improve our manuscript.

**General comments**
**This well-written and well-considered manuscript set out to explore the variability of stream DOC in both burned and unburned parts of a sub-catchment. The paper builds on a large dataset of streamflow DOC and uses satellite-derived burn severity to explore how burn severity impacts stream DOC at seasonal scales. The main weakness of the paper is that data collection was delayed by > 2 years post-fire. Greater consideration is needed for the delayed onset of sampling, and how this may have altered results. The main strength is that the dataset is extensive (129 sites, repeated collection), allowing for interesting and useful statistics. The paper was an enjoyable read, and I look forward to seeing the revised version.**

- Thank you for your comment. It is fairly typical of post-fire research to have delayed sampling, often due to limitations to site access. We agree with the reviewer that greater consideration for the fact that we sampled two years post-fire is an area that could be further explored in terms of potential limitations or influence on the results in the manuscript. We have seen evidence from past research (i.e., Rhoades et al. 2019; Emelko et al., 2016; Niemeyer et al. 2020) that wildfire effects on hydrology and water quality can persist for more than a decade. Additionally, several meta-analyses have found that DOC impacts lasted at least 5 years (Cavaiani et al. 2024; Raoelison et al., 2023; Hampton et al., 2022; Rust et al., 2018)--thus, it was our expectation that the effects from the high severity wildfire in our study would result in substantial long-term effects on water quality. P4, L94-100: "For example, DOC concentrations may be affected by revegetation and recovery of landscapes over time after wildfire. While some have estimated recovery as short as seven months (Wei et al., 2021), several studies have illustrated that wildfire effects may persist for 10 or more years (Chow et al., 2019; Santos et al., 2019; Parham et al., 2013; Rodríguez-Cardona et al., 2020). Moreover, results from several meta-analyses have shown that impacts from wildfire on DOC concentrations may persist for at least five years (Cavaiani et al., 2024; Raoelison et al., 2023; Hampton et al., 2022; Rust et al., 2019). Post-fire DOC concentrations can also be dependent on area burned (Rhoades et al., 2019; Uzun et al., 2020; Chow et al., 2019) and fire severity (Santos et al., 2019)."
- P22, L421-429: we added the following text to discuss this limitation "Lastly, the lack of signal could be due to the timing of sampling, since we started sample collection two years post-fire. Past research has found that the first few storms post-fire can be critical periods of flushing of ash and wildfire debris from the landscape, often leading to the most substantial increases in DOC concentrations (Writer et al., 2012). However, this was not necessarily observed following the Holiday Farm fire. A previous study in the McKenzie River sub- basin quantified DOC concentrations between 1.5 to 3.0 mg L$^{-1}$ across five burned tributaries during the first major storm post-fire (Roebuck Jr. et al., 2022), which is similar to the concentrations we observed (Fig. 4), suggesting a similar DOC response to the storm., Indeed, results from a recent meta-analysis did not illustrate relationship between the time between the wildfire, water sampling, and post-fire DOC concentrations (Raoelison et al., 2023)."

**Ordinarily, I would have reviewed the code, but I saw too late that it was available to the reviewers on the same page were reviewer comments are posted. My apologies that I did not see it in time, and cannot comment on it here.**

**Specific comments**
**Without any pre-fire data, some statements become tenuous: e.g. Line 382: Wildfires are known to impact organic matter availability on burned hillslopes and shift hydrologic flow paths… we expected DOC**

concentrations would be influence…" but you are analysing a post-fire system, in which DOC is already low and buffered by the streamflow. So perhaps you're too late for the post-fire DOC signal/it is too subtle/it is being confounded by the upstream-downstream gradient?

- After examining the initial spatial patterns, we used the spatial stream network models to investigate if the signal was too subtle to be observed spatially and if it was being confounded by landscape factors (Figures 6 & 7). In terms of a lack of pre-fire data, while we don't have pre-fire data, we do have 65 unburned reference control sites across the basin which help us isolate the impact of the wildfire.
- In terms of being too late for the post-fire DOC signal, it is possible that we were too late to observe the post-fire signal, however current wildfire recovery literature suggests that impacts last 5+ years (Cavaiani et al. 2024). To better address this we added the following text P4, L94-100: "For example, DOC concentrations may be affected by revegetation and recovery of landscapes over time after wildfire. While some have estimated recovery as short as seven months (Wei et al., 2021), several studies have illustrated that wildfire effects may persist for 10 or more years (Chow et al., 2019; Santos et al., 2019; Parham et al., 2013; Rodríguez-Cardona et al., 2020). Moreover, results from several meta-analyses have shown that impacts from wildfire on DOC concentrations may persist for at least five years (Cavaiani et al., 2024; Raoelison et al., 2023; Hampton et al., 2022; Rust et al., 2019). Post-fire DOC concentrations can also be dependent on area burned (Rhoades et al., 2019; Uzun et al., 2020; Chow et al., 2019) and fire severity (Santos et al., 2019)."
- P22, L421-429: we added the following text to discuss this limitation "Lastly, the lack of signal could be due to the timing of sampling, since we started sample collection two years post-fire. Past research has found that the first few storms post-fire can be critical periods of flushing of ash and wildfire debris from the landscape, often leading to the most substantial increases in DOC concentrations (Writer et al., 2012). However, this was not necessarily observed following the Holiday Farm fire. A previous study in the McKenzie River sub- basin quantified DOC concentrations between 1.5 to 3.0 mg L$^{-1}$ across five burned tributaries during the first major storm post-fire (Roebuck Jr. et al., 2022), which is similar to the concentrations we observed (Fig. 4), suggesting a similar DOC response to the storm., Indeed, results from a recent meta-analysis did not illustrate relationship between the time between the wildfire, water sampling, and post-fire DOC concentrations (Raoelison et al., 2023)."

Is the burn severity in the model a mean burn severity for the whole fire, or the associated burn severity for each collection point? Since the heterogeneity was likely higher than what is resolveable by the satellite products, you could stress-test how you select burn severity, by using different mean burn severities within different radii of each collection point.

- Thank you for catching this, we did not explain how we determined burn severity very clearly. It was determined as the average burn severity (determined by dNBR) across the upstream area for each sampling point. We did try other methods (i.e,. using the average dNBR for a 100m buffer on either side of the stream, the percentage burned a high severity) but these did not notably influence the results.
- P8 L146-148: We added the text: "In particular, burn severity was calculated as the average dNBR value across each site's upstream contributing area, with unburned areas counted as dNBR value of 0." To clarify how burn severity was determined.

The discussion refers to hypotheses (Line 473) which is not clearly stated in the introduction. Please make sure to clearly state any hypotheses in the introduction, or just refer to your clearly defined research questions. This section continues to say that "Hydrologic flow paths would shift to more shallow pathways" during the wet season – is this an over-simplification? Would it instead be that during the wet season the proportion of hydrological flow from more shallow pathways is greater, due to surface runoff and lag in infiltration? Shift implies to me that the contribution from groundwater declines. Suggest rewording to make this clearer.

- As you stated, we would expect that a greater overall proportion of streamflow would come from lateral flow during the wet season (our basins have extremely limited surface runoff). You make a good point that the use of the word "shift" is misleading, we clarified this section to remove reference to hypotheses, link to our existing research questions defined in the introduction, and clarify our ideas.
- P25 L503-506: We altered the text to read : "In previous work, researchers have shown that DOC is flushed through shallower flow paths during wet periods and is more connected to the landscape than during dry

periods (Tiwari et al., 2014). As such, we expected the largest wildfire impacts during the wet season when the burned hillslopes were most connected to the streams."

**Line 245: "contrary to expectations, we did not observe…" The fire was in September 2020 but the first sampling happened in November 2022. How long would you expect a signal to persist? The introduction needs to refer to the literature on the persistence of post-fire changes in DOC, so we can understand the potential for the post-fire DOC signal to persist in the landscape.**

- We agree that this is an area we could further explore and in revision we will include more discussion on post-fire recovery. Past work (i.e. Rhoades et al. 2019) measured fire impacts 14 years post-fire in Colorado while a recent meta analysis found that DOC impacts lasted at least 5 years across North America (Cavaiani et al. 2024). Emelko et al., (2011) quantified elevated DOC >10 years after wildfire in the Rocky Mountains. Niemeyer et al. (2020) was able to identify elevated streamflow >30 years after a wildfire. These are only a few examples–there is evidence that wildfires are a substantial perturbation to the system, which can create effects that persist for decades. As such, it is our expectation that we would observe an effect of wildfire just two years after.
- To better address this we added the following text P4, L94-100: "For example, DOC concentrations may be affected by revegetation and recovery of landscapes over time after wildfire. While some have estimated recovery as short as seven months (Wei et al., 2021), several studies have illustrated that wildfire effects may persist for 10 or more years (Chow et al., 2019; Santos et al., 2019; Parham et al., 2013; Rodríguez-Cardona et al., 2020). Moreover, results from several meta-analyses have shown that impacts from wildfire on DOC concentrations may persist for at least five years (Cavaiani et al., 2024; Raoelison et al., 2023; Hampton et al., 2022; Rust et al., 2019). Post-fire DOC concentrations can also be dependent on area burned (Rhoades et al., 2019; Uzun et al., 2020; Chow et al., 2019) and fire severity (Santos et al., 2019)."
- P22, L421-429: we added the following text to discuss this limitation "Lastly, the lack of signal could be due to the timing of sampling, since we started sample collection two years post-fire. Past research has found that the first few storms post-fire can be critical periods of flushing of ash and wildfire debris from the landscape, often leading to the most substantial increases in DOC concentrations (Writer et al., 2012). However, this was not necessarily observed following the Holiday Farm fire. A previous study in the McKenzie River sub- basin quantified DOC concentrations between 1.5 to 3.0 mg $L^{-1}$ across five burned tributaries during the first major storm post-fire (Roebuck Jr. et al., 2022), which is similar to the concentrations we observed (Fig. 4), suggesting a similar DOC response to the storm., Indeed, results from a recent meta-analysis did not illustrate relationship between the time between the wildfire, water sampling, and post-fire DOC concentrations (Raoelison et al., 2023)."

**The consideration of drinking water feels tacked on in the conclusion. As you did not do any characterisation of which compounds make up your DOC, talking about effects on DOC if more recalcitrant types of DOC are formed comes out of nowhere. The discussion does not mention impacts on drinking water at all, and it is only a minor part of the introduction. If this is included to set up further work, it should either be presented more concisely, or the discussion should be expanded to include drinking water and how your results relate to drinking water.**

- Our aim is to set up future work on DOM character with the conclusion. However, we agree it could be presented more concisely.
- P26, L545-557: Conclusion was re-written to read "While our results provided additional context for the wide variability of post-fire DOC responses reported in the literature, a universal understanding of the response of DOC to wildfire remains unresolved. For example, while we observed little shifts in DOC concentrations following wildfire, our study did not address the potential changes in dissolved organic matter character that may occur, which can influence its fate in the environment. Thus, further work exploring how DOC concentrations and DOM character changes with burn severity across both space and time could further improve our understanding of the mechanisms of delivery of DOC from burned hillslopes to streams (Roebuck Jr. et al., 2022, 2023). This type of research is necessary to improve mechanistic representation of DOC and DOM character in models to facilitate important post-fire predictions of the likely range of response. However, our work also highlights the need to consider a broad range of potentially confounding landscape factors that can influence the hydrobiogeochemical response to wildfire."

**Technical corrections**

**Line 15: just state the number of sites rather than ~**

- We updated the manuscript with the total number of sites sampled (129) instead of using an estimated number in the abstract as suggested
- P1, L15: We changed to text from "~100" to 129

**Line 141: how was severity classified (briefly)?**

- Burn severity is determined by finding the satellite derived difference in normalized burn ratio (dNBR) from the pre to post-fire period. Burn severity classes are determined by examining thresholds in the data. We added these details into the manuscript.
- P8, L146-148: We added the text "Monitoring trends in burn severity (MTBS) burn severity is determined from the difference in satellite derived normalized burn ratio (dNBR) from the pre- to post-fire period, with the burn severity classifications assigned by based on thresholds in the data."

**Figure 1: data sources should be cited within the caption.**

- P7, L135: We altered the caption to include the geospatial data citations. "Figure 1: (a) Map of the McKenzie River sub-basin, Oregon USA (U.S. Geological Survey, 2020) and associated land uses (Dewitz, 2021). Sub-basins of particular interest and USGS gauges used in Figure 2 are labeled. (b) Map of the burn severity of the 2020 Holiday Farm wildfire (MTBS Project, 2021) and water sampling sites distributed across the stream network in the McKenzie sub-basin. The shapes indicate the location of each site relative to the perimeter of the Holiday Farm wildfire."

**Figure 2: define the water year (i.e. from which bracketing months?). The USGS data should have a reference. What synoptic sampling, there are no precipitation data presented? Either here on in an extended figure in the supplement, you should show precip and hydrographs for the 2021 and 2022 water years (with the timing of the fire shown).**

- P9, L163-168: We altered the caption of Figure 2 to read (a) Daily precipitation from Oct 2022 to Oct 2023 in the McKenzie River sub-basin (44.2119, -122.2559) (Daly, 2023). (b) Discharge at three USGS stream gauges throughout the McKenzie sub-basin from Oct 2022 to Oct 2023 showing discharge patterns for the lower (Camp Creek), middle (Gate Creek), and upper (McKenzie River at Outlet of Clear Lake) regions of the sub-basin. Dates of water sample collection are labeled with vertical dashed lines. Data was obtained using the dataRetrieval package in R (Cicco et al., 2018; U.S. Geological Survey, n.d.)."
- We replaced the words water year with streamflow from Oct. 2022 to Oct. 2023 to be clearer. We also added the reference of the USGS data to the figure caption.
- We were confused about your comments that "there are no precipitation data presented", as precipitation is shown in Figure 2a.
- The synoptic sampling refers to the four sampling campaigns; we altered the caption to change synoptic sampling to water sample collection to be clearer.
- We feel that including an extended figure of precipitation and the hydrographs is outside the scope of our study since we don't deal with those time periods at all.

**Line 167: move the lines from 172 about need to freeze samples for DOC up to line 166, as justification for why samples weren't frozen.**

- We agree the movement of those lines makes more sense as suggested, we moved the lines as suggested.

**Line 173: Change 'doesn't' do 'does not'**

- We changed this word to remove the contraction.

**Line 201: states seasonal models used same variables as mean models, but line 215 says season was not included in the seasonal models (which is sensible, but need to be clear about what independent variables are used where).**

- Thanks for pointing this out, the seasonal models did not include season. We altered the text in line 201 to read P11, L216-217 "…except for season, as this was constant in each seasonal model."

**Line 206: since you only used a tail-up model, is this necessary?**

- We think including explanations of the other types of models is necessary since, while we ended up just using a tail-up model, we tested all three types so it provides context for our results.

**Line 215: suggest rewording "checked for model issues" to something like "ensure that the assumptions of linearity, normality, and homoscedasticity were met" (if that is correct)**

- Thank you for your suggestion, we agree we could be more clear. We checked our models by examining the residuals and performing leave one out cross validation to ensure that the assumptions of linearity, normality, and homoscedasticity were met.
- P12, L231-232: We altered the text to read "we checked our models by examining the residuals and performing leave one out cross validation to ensure that the assumptions of linearity, normality, and homoscedasticity were met."

**Line 220: What is the reference for thresholds? Common source of thresholds is Key and Benson (2006), who describe different threshold values to those used here.**

- On line 236 we state the thresholds used were "based on the dNBR thresholds monitoring trends in burn severity used in classifying the burn severity for the Holiday Farm Fire (MTBS Project, 2021)."
- P12, L236-237: We altered the text to read "For descriptive statistics we chose to bin the continuous burn severity dNBR values into "unburned", "low", "moderate", or "high" based on the dNBR burn severity thresholds determined by MTBS for the Holiday Farm Fire specifically monitoring trends in burn severity used in classifying the burn severity for the Holiday Farm Fire (MTBS Project, 2021)."

**Line 266 – how were CIs calculated?**

- CI were calculated using the standard errors from the GLMM model. We edited the methods section to include a more thorough description.
- P13, L 220: We added the text "Confidence intervals for the mean and variance were calculated using the standard errors from model."

**Line 295: define 'nugget.' It is used in the text before it is defined in the caption for Table 3.**

- You are correct, thank you for catching that, we included a definition of the nugget (the unexplained variance in the SSN model) where it is first used.
- P17, L 313-314: We altered the sentence referenced to read: "Despite this, the mean model had a large nugget (38.0 %), or the unexplained variance not explained by the covariates and spatial autocorrelation, suggesting there is quite a bit of variance unaccounted for in the model."

**Line 298 – a caveat should be added in here somewhere about the large standard error of the coefficient values for dNBR, AI, and pH, noting overlap with most other variables.**

- P17 L317-318: You make a good point, we will add in text to emphasize the overlapping confidence intervals by adding the text: "However, the confidence intervals for these variables overlap, suggesting uncertainty in the exact order of importance for these variables."

**Figure 3: predicted error is not resolvable at this scale. Suggest plotting the predicted error elsewhere and moving to the supp mat. I don't think a diverging colour palette is the right choice for this dataset.**

- We tried a number of different color palettes for this dataset, the one used was the only one we could find that was colorblind friendly and still allowed for interpretation of the results accurately. If you have specific color palette suggestions, we'd happily try them.
- Figure 3: We removed the predicted error and plotted the points using a uniform size. As suggested, we created a supplemental figure (Figure A3) where the size differences between the points is larger to be more easily interpretable for the standard error.

**Line 363: introducing 'burn severity thresholds determined by MTBS' here sounds like a different burn severity index than the dNBR used earlier in the manuscript. From the methods, it looks like there is only one product used? Please clarify.**

- P21, L381-383: We altered the sentence to read "To better contextualize our results, we used the burn severity thresholds determined by MTBS for the Holiday Farm fire to estimated the average change in DOC across each MTBS burn severity group from using the coefficients from our SSN models (Fig. 7)." To clarify.

**Figure 7: if possible, could you rearrange either figure 7 or figure 4 so that they match (e.g. colour = burn severity, x axis has antecedent conditions, or vice versa) to allow for better comparison between model and obs.**

- We rearranged Figure 7 to match the colors/axis of figure 4. However, to clarify, these two figures shouldn't necessarily be compared, they are not presenting the same thing. Figure 4 is the overall DOC across season and severity levels. Figure 7 is presenting the predicted *change* in DOC only due to *wildfire* (removing other confounding factors).

- We edited the caption of Figure 7 to make this clearer changing the text to "The change in dissolved organic carbon (DOC) predicted by each of the seasonal spatial stream network models at each burn severity level for the McKenzie River sub-basin, OR. Predictions were based on the burn severity coefficient in the fitted spatial stream network (SSN) models. The central box illustrates the range in change based on the low and high threshold values for each burn severity group. The error bars are based on the standard error of the burn severity coefficient, where the upper bar is the upper 95 % confidence interval for the low threshold values and the lower bar is the lower 95 % confidence interval for the high threshold values.".

**Line 434: suggest using 'found' rather than 'measured'**

- We accept this change in word choice from the reviewer.
- P23, L463: Changed the word "measured" to "found"

**Line 436: missing word? Following a wildfire in Alberta?**

- P23, L466: Thank you for catching that, the reviewer's suggestion is correct, there was some text missing in that sentence. We modified the sentence to read: "The importance of groundwater on DOC concentrations after wildfire was also noted following a wildfire in Alberta, CAN, where there was no measurable impact of wildfire on DOC concentrations in fens (Davidson et al., 2019) or boreal lakes (Olefeldt et al., 2013)."

**Line 441: refs should be together in parentheses.**

- P23, L 469: Thank you for catching that, those references should be together. We merged the citation together into one.

**Line 491: Please clarify, increased contributions of DOC from groundwater, or greater contribution of groundwater flow?**

- P25, L524-526: Thanks for noting this, that is unclear. They noted increased contributions of groundwater to overall streamflow. We rephrased this sentence to read: "In recent work, authors have reported that wildfire can lead to increased groundwater contributions to overall streamflow (MacNeille et al., 2020; Rey et al., 2023)—others doing previous research in the same basin identified the lowest DOC concentration in the areas with the greatest groundwater inputs (Kraus et al., 2010).

**Line 492: please rephrase, the 'while' sounds like the second clause will disagree, but the second clause supports your statement.**

- P25, L524-526: Thanks for noting this, that is unclear. They noted increased contributions of groundwater to overall streamflow. We rephrased this sentence to read: "In recent work, authors have reported that wildfire can lead to increased groundwater contributions to overall streamflow (MacNeille et al., 2020; Rey et al., 2023)—others doing previous research in the same basin identified the lowest DOC concentration in the areas with the greatest groundwater inputs (Kraus et al., 2010).

**Line 763: Ref should state that this is a preprint.**

- P41, L812-813: Yes, that was an oversight, we updated this citation.

**Line 807: Add URL.**

- P42, L853: We added the URL for this citation.

**REVIEW 2**

**This is a review of "The influence of burn severity on dissolved organic carbon concentrations across a stream network differs based on seasonal wetness conditions post-fire" by Wampler et al.**

- Thank you for taking the time to provide thoughtful and helpful comments to improve our manuscript.

**The manuscript analyzes spatial and temporal dynamics of stream DOC concentration in a watershed impacted by extensive wildfire about two years prior to the sampling campaign. The extensive spatial sampling (about 100 sites) allows the inference of landscape attributes affecting DOC concentration and how they vary with watershed wetness condition. The focus variable of the authors, burn severity, has a minor role in predicting DOC concentration.**

**Overall, the manuscript is very well written, organized and easy to follow. The methods are clear (excepts for few points detailed below) and sound. My few comments are reported below.**

**The first one is of general character. One obvious limitation of the study is that it comprises samples taken only after the fire. I think that the introduction would benefit from classifying the existing literature depending on whether before/after data were available. This kind of limitation should also surface in the discussion/conclusion. Another potential issue is related to the time elapsed between the wildfire and the data collection. The literature review should discuss more in details the expected duration of the potential impact.**

- Thank you for your comment. It is fairly typical of post-fire research to have delayed sampling, often due to limitations to site access. We agree with the reviewer that greater consideration for the fact that we sampled two years post-fire is an area that could be further explored in terms of potential limitations or influence on the results in the manuscript. We have seen evidence from past research (i.e., Rhoades et al. 2019; Emelko et al., 2016; Niemeyer et al. 2020) that wildfire effects on hydrology and water quality can persist for more than a decade. Additionally, several meta-analyses have found that DOC impacts lasted at least 5 years (Cavaiani et al. 2024; Raoelison et al., 2023; Hampton et al., 2022; Rust et al., 2018)--thus, it was our expectation that the effects from the high severity wildfire in our study would result in substantial long-term effects on water quality. To better address this we added the following text P4, L94-100: "For example, DOC concentrations may be affected by revegetation and recovery of landscapes over time after wildfire. While some have estimated recovery as short as seven months (Wei et al., 2021), several studies have illustrated that wildfire effects may persist for 10 or more years (Chow et al., 2019; Santos et al., 2019; Parham et al., 2013; Rodríguez-Cardona et al., 2020). Moreover, results from several meta-analyses have shown that impacts from wildfire on DOC concentrations may persist for at least five years (Cavaiani et al., 2024; Raoelison et al., 2023; Hampton et al., 2022; Rust et al., 2019). Post-fire DOC concentrations can also be dependent on area burned (Rhoades et al., 2019; Uzun et al., 2020; Chow et al., 2019) and fire severity (Santos et al., 2019)."
- P22, L421-429: we added the following text to discuss this limitation "Lastly, the lack of signal could be due to the timing of sampling, since we started sample collection two years post-fire. Past research has found that the first few storms post-fire can be critical periods of flushing of ash and wildfire debris from the landscape, often leading to the most substantial increases in DOC concentrations (Writer et al., 2012). However, this was not necessarily observed following the Holiday Farm fire. A previous study in the McKenzie River sub- basin quantified DOC concentrations between 1.5 to 3.0 mg L$^{-1}$ across five burned tributaries during the first major storm post-fire (Roebuck Jr. et al., 2022), which is similar to the concentrations we observed (Fig. 4), suggesting a similar DOC response to the storm., Indeed, results from a recent meta-analysis did not illustrate relationship between the time between the wildfire, water sampling, and post-fire DOC concentrations (Raoelison et al., 2023)."

**Lines 195-200. Could you please expand on the rationale for the choice of this variable selection procedure? I think I have intuitively understood it, but I would suggest to be more explicit.**

- P10, L209-211: We chose to use the double selection procedure because it is robust, allows more accurate identification of potential confounding variables. Most importantly, the method prevents inflation of the p-values and standard errors for our variable of interest, burn severity. We adding the following text to justify the choice: "We chose to use the double selection procedure because it is robust, allowing more accurate identification of potential confounding variables. Most importantly, the method prevents inflation of p-values and standard errors for our variable of interest, burn severity (Belloni et al., 2014)."

**Line 381-387 and 506-510. This is a fair account of the results, but I feel that is not effectively summarized in the title. I can understand that the authors are attached to their initial hypothesis, but maybe they could consider changing it.**
- One of the key main points of our paper is that the importance of burn severity isn't constant, but changes across the seasons. Given this, we feel that the title does convey our results, even if we're not able to incorporate all our main findings into the title. We did remove the words "post-fire" from the title as this is redundant as we've already mentioned burn severity.

**Line 448. Maybe it is worth noting here that the theoretical expectation for a uniform stream network is that DOC concentration decreases with drainage area due to instream removal (se e.g. https://doi.org/10.1016/j.advwatres.2017.10.009)**
- Indeed, as you pointed out we would expect more instream removal leading to lower DOC concentrations at higher stream orders. We included a statement of this and added the reference from the reviewer in our discussion.
- P24 L474-476: We altered the text to read: "This is consistent with our conceptual understanding of DOC through a stream network, where DOC decreases with drainage area due to in-stream removal (Bertuzzo et al., 2017). This pattern of homogenization has also been noted by others (Bhattacharya and Osburn, 2020; Creed et al., 2015) who similarly attributed the lower variability in DOC concentrations at high stream orders to hydrological averaging and a dominance of in-stream processes."

**MINOR COMMENTS**
**Table 1. For most variables, it is explicitly reported that the values consider the whole basin area upstream of the point. For the soil variables and the TWI this is not explicitly stated. Please clarify.**
- Table 1: Thanks for catching this, soil variables and TWI were averaged over the upstream area. We updated the text to reflect this by adding the text "… averaged across the upstream basin area."

**Figure 2. Please report the location of the three outlets in Figure 1.**
- Thanks for the suggestion, that's a great idea. We edited Figure 1 as you suggested to include the location of the USGS gauges used in Figure 2.

**Line 255 (and other places). I think you can omit "OR" after the first occurrence.**
- Thanks for the suggestion, we added it in so that if someone is just scanning through the paper and hasn't read the methods section, they have the information needed to interpret the figure.

**Line 394. Delete "rates of". Hydraulic conductivity is not a rate.**
- P22, L414,416: Saturated hydraulic conductivity, $K_{sat}$, describes water movement through saturated media, and its units are distance/time. Therefore, we believe our sentence is scientifically correct. However, we recognize that there are many different similar concepts related to $K_{sat}$ that might cause confusion for readers. Therefore, we included $K_{sat}$ into the sentence to increase clarity on what specific parameter the hydraulic conductivity is referencing in this sentence.